# VITA: Zero-Shot Value Functions via Test-Time Adaptation of Vision–Language Models

**Christos Ziakas**[*]
Department of Computing
Imperial College London

**Alessandra Russo**
Department of Computing
Imperial College London

## Abstract

Vision–Language Models (VLMs) show promise as zero-shot goal-conditioned value functions, but their frozen pre-trained representations limit generalization and temporal reasoning. We introduce VITA, a zero-shot value function learning method that enhances both capabilities via test-time adaptation. At inference, a lightweight adaptation module is updated via a gradient step on a meta-learned self-supervised loss, such that each test-time update improves value estimation. By updating sequentially over a trajectory, VITA encodes history into its parameters, addressing the temporal reasoning limitations. To mitigate shortcut learning, we propose a dissimilarity-based sampling strategy that selects semantically diverse segments of the trajectory during training. In real-world robotic manipulation tasks, VITA generalizes from a single training environment to diverse out-of-distribution tasks, environments, and embodiments, outperforming the state-of-the-art zero-shot method using autoregressive VLMs. Furthermore, we demonstrate that VITA's zero-shot value estimates can be utilized for reward shaping in offline reinforcement learning, resulting in multi-task policies on the Meta-World benchmark that exceed the performance of those trained with the simulation's fuzzy-logic dense rewards. Project website: `https://chziakas.github.io/vita/`

## 1 Introduction

Vision-Language Models (VLMs) have demonstrated strong generalization across diverse tasks and domains by learning from large-scale, unstructured web data without human supervision (Radford et al., 2021; Alayrac et al., 2022). In contrast, despite significant advances in learning generalist policies for robotics (Brohan et al., 2023; Kim et al., 2024) and 3D virtual environments (Reed et al., 2022; SIMA Team et al., 2024), state-of-the-art methods have yet to achieve comparable success due to their reliance on expert demonstrations (Ho & Ermon, 2016). World models (Ha & Schmidhuber, 2018; Hafner et al., 2021; Matsuo et al., 2022) have emerged as a promising solution, enabling agents to learn in simulated environments, with applications spanning robotics (Yang et al., 2023; Agarwal et al., 2025), autonomous driving (Hu et al., 2024), and video games (Bruce et al., 2024). Despite the potential of world models to generate realistic visual trajectories in diverse, open-ended environments (Hughes et al., 2024; Silver & Sutton, 2025), a critical challenge remains: *how can agents effectively learn from videos at scale without relying on human supervision?*

A line of research explores policy learning directly from expert visual trajectories by inferring actions from visual transitions (Edwards et al., 2019; Baker et al., 2022; Schmidt & Jiang, 2024; Zhang et al., 2025a). These latent actions can be mapped to executable controls in deployment, although this remains a challenging task in practice. In parallel, other work focuses on learning a universal *goal-conditioned value function*, using it as a zero-shot reward shaping and supervision signal for reinforcement learning (RL) and imitation learning, respectively (Chen et al., 2021; Ma et al., 2022). In this framework, goal-conditioned value estimation can be formulated as: predicting how far an agent has progressed toward completing a task, based on visual observations and a natural language task description (Lee et al., 2021a; Ma et al., 2023b; 2024; Dashora et al., 2025).

Prior work has employed pre-trained contrastive VLMs (Radford et al., 2021) for zero-shot reward shaping, leveraging the similarity between task descriptions and visual observations in a shared

---

[*]Corresponding author: `c.ziakas24@imperial.ac.uk`

multimodal representation space to ground progress in semantic context (Ma et al., 2023b; Baumli et al., 2023; Rocamonde et al., 2024). However, these methods fail to capture the temporal context needed to disambiguate visually similar states that occur at different stages of a task (e.g., folding vs. unfolding a shirt). In contrast, autoregressive VLMs (Alayrac et al., 2022) incorporate temporal context by conditioning on the entire visual trajectory within the prompt; however, they inherit a bias toward monotonically increasing predictions from the chronologically ordered datasets used in pre-training (Ma et al., 2024). Both VLM architectures rely on pre-trained representations (Ma et al., 2023b; 2024; Baumli et al., 2023; Rocamonde et al., 2024; Du et al., 2023) for zero-shot prediction, which limits both their generalization and their temporal reasoning (Pătrăucean et al., 2023). To enable generalizable value function learning, recent work has explored large-scale pre-training (Alakuijala et al., 2024; Ma et al., 2023b), domain-specific fine-tuning (Zhang et al., 2025b), and demo-based reward specification (Sontakke et al., 2023).

In this work, we introduce VITA, a zero-shot value function learning method that enhances both generalization and temporal reasoning through test-time adaptation, outperforming the state-of-the-art zero-shot approach (Ma et al., 2024) on real-world robotic manipulation tasks. Unlike prior methods that rely on large-scale pre-training or expert demonstrations for generalization, VITA adapts online to both the semantic and temporal context of trajectories. At inference, a lightweight adaptation module is updated in negligible time with a gradient step on a meta-learned self-supervised loss (Sun et al., 2024), such that each test-time update improves value function estimation (Finn et al., 2017). By updating sequentially over a test trajectory, the value function estimator encodes trajectory history into its parameters (Sun et al., 2024), thereby addressing the temporal reasoning limitations of prior work. To mitigate shortcut learning (Geirhos et al., 2020), we introduce a dissimilarity-based sampling strategy that encourages reliance on semantic cues, supported by empirical evidence. Our evaluation demonstrates the effectiveness of our method across core capabilities of value function estimation: generalization under distribution shifts, differentiation between expert and non-expert trajectories, and reward shaping for offline RL (Levine et al., 2020).

We summarize our key contributions as follows:

- We propose VITA, a test-time adaptation method that enhances the generalization and temporal reasoning of contrastive VLMs for zero-shot value function estimation, without requiring task-specific demonstrations or large-scale pre-training.

- VITA generalizes from a single training environment to diverse out-of-distribution tasks, environments, and embodiments in robotic manipulation, outperforming the state-of-the-art zero-shot method (Ma et al., 2024).

- VITA's zero-shot value estimates for reward shaping yield offline RL policies on the Meta-World benchmark for multi-task learning (MT10) that surpass those trained with the simulation's fuzzy-logic dense rewards (Yu et al., 2020).

## 2 PRELIMINARIES

### 2.1 GOAL-CONDITIONED VALUE FUNCTIONS

In video trajectories, task progress estimation can be viewed as goal-conditioned value function estimation, where the reward reflects task completion. Therefore, we formulate learning a goal-conditioned value function as predicting the degree of task completion from vision–language representations. Formally, the value function is defined as: $V : \mathcal{O} \times \mathcal{G} \to [0, 1]$ which maps an observation $o_t \in \mathcal{O}$ and a goal specification $g \in \mathcal{G}$ to a scalar value indicating the predicted progress toward goal completion. We set $V(o_t; g) = 0$ to correspond to the start of the task and $V(o_t; g) = 1$ to indicate completion. Task progress is commonly aligned with temporal position in expert demonstrations, based on the assumption that such trajectories exhibit monotonically increasing progress toward goal completion (Lee et al., 2021a; Ma et al., 2024; Dashora et al., 2025). Given an expert trajectory $\tau = (o_1, \ldots, o_T) \sim \pi_E$, temporal progress is defined using normalized timestep indices: $V^{\pi_E}(o_t; g) = \frac{t}{T}$ where $T$ is the trajectory length. Therefore, temporal progress provides supervision for learning a goal-conditioned value function $V$.

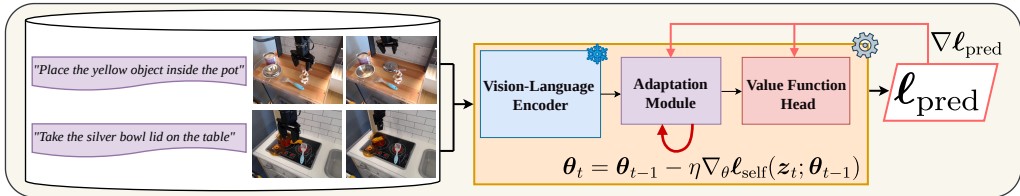

(a) Training: Learning goal-conditioned value function estimator via meta-learning

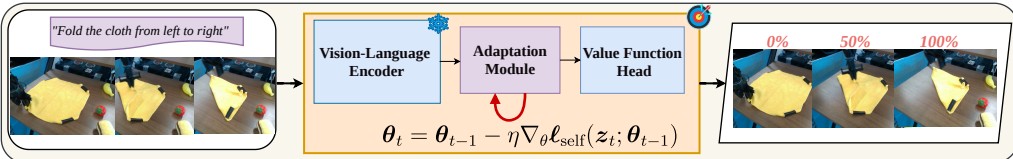

(b) Inference: Zero-shot value estimate for OOD test trajectory via test-time adaptation

Figure 1: **Overview of VITA.** VITA learns a goal-conditioned value function via meta-learning and achieves zero-shot generalization to out-of-distribution trajectories via test-time adaptation.

## 2.2 TEST-TIME TRAINING FOR SEQUENCE MODELING

*Test-time training* (TTT) (Sun et al., 2020) is a test-time adaptation method that treats inference as a self-supervised learning task (Goodfellow et al., 2016), updating model parameters on each test instance without access to labels. While originally proposed for static image classification (Sun et al., 2020), recent work extends TTT to sequence modeling (Sun et al., 2024; Wang et al., 2023), where a parametric adaptation module $f_{\text{adapt}}$ is updated at each timestep using a gradient step on a self-supervised loss. In contrast to standard sequence architectures that encode temporal history in activation states, such as in hidden states (Hochreiter & Schmidhuber, 1997; Cho et al., 2014) or in cached key–value pairs (Vaswani et al., 2017), TTT encodes history into the parameters of $f_{\text{adapt}}$ via sequential updates in inference. The parameters of $f_{\text{adapt}}$ thereby serve as an implicit memory, encoding temporal history while preserving temporal order. Sun et al. (2024) further propose to meta-learn the self-supervised task, following the gradient-based meta-learning paradigm (Finn et al., 2017). In this setting (Sun et al., 2024), the self-supervised loss is parameterized by learnable linear projections and trained to minimize downstream prediction loss after a test-time update, rather than directly minimizing it. In this work, we apply TTT to goal-conditioned value function learning, where temporal context is captured through sequential updates, and the self-supervised task is meta-learned to improve value estimation rather than being predefined a priori.

## 3 VITA: ZERO-SHOT VALUE FUNCTIONS VIA TEST-TIME ADAPTATION

### 3.1 MODEL ARCHITECTURE

Our goal-conditioned value function estimator comprises three modules: (1) a multimodal encoder that extracts joint visual-language representations from visual trajectories and their task descriptions; (2) an adaptation module updated at test-time using a self-supervised loss that is meta-learned to improve value estimation; and (3) a regression head that predicts value estimates.

### 3.1.1 MULTIMODAL INPUT REPRESENTATION

We use a frozen contrastive vision-language encoder, CLIP (Radford et al., 2021), to extract representations from visual observations and goal descriptions. Given a visual trajectory $\tau = (o_1, \ldots, o_T)$ and a language task description $g$, we concatenate their representations at each timestep $t$ to form a sequence of joint multimodal representations $(z_1, z_2, \ldots, z_T)$, where $z_t = [\phi_v(o_t); \phi_g(g)] \in \mathbb{R}^{2d}$, and $\phi_v$ and $\phi_g$ denote the visual and language encoders, respectively. CLIP is pre-trained with a contrastive objective to align paired image-text inputs in a shared representation space (Radford et al., 2021). As a result, representations of visual observations that are semantically closer to goal completion tend to be closer to the representations of the goal description.

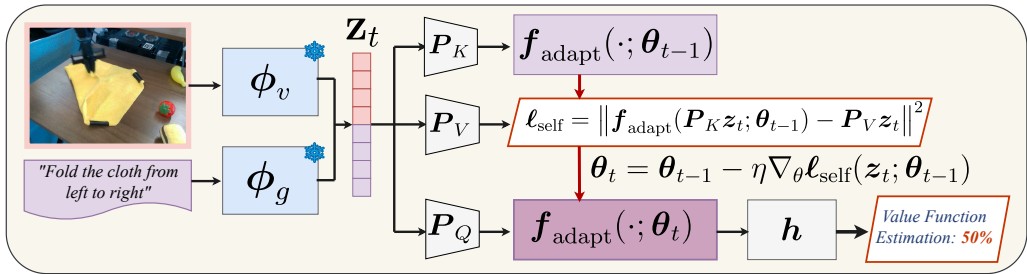

Figure 2: **Test-time Adaptation.** In inference, at each timestep $t$, an adaptation module $f_{\text{adapt}}$ is updated via a gradient step on a meta-learned self-supervised loss $\ell_{\text{self}}$, encoding temporal history.

### 3.1.2 TEST-TIME ADAPTATION

To adapt multimodal representations to both semantic and temporal context, we employ an adaptation module $f_{\text{adapt}}$ following the test-time training paradigm described in 2.2. At inference, the parameters of $f_{\text{adapt}}$ are updated online via a gradient step on a meta-learned self-supervised loss $\ell_{\text{self}}$. This loss is formulated as a reconstruction objective using learnable linear projections $P_K$ and $P_V$, which are meta-learned during training to improve value estimation after test-time adaptation. In particular, given $P_K \in \mathbb{R}^{d' \times 2d}$ that generates a perturbed input view and $P_V \in \mathbb{R}^{d' \times 2d}$ for the target, the self-supervised loss is defined as:

$$\ell_{\text{self}}(z_t; \theta_{t-1}, P_K, P_V) = \|f_{\text{adapt}}(P_K z_t; \theta_{t-1}) - P_V z_t\|^2. \tag{1}$$

where $\theta_{t-1}$ denotes the parameters of $f_{\text{adapt}}$ before test-time adaptation. At each timestep $t$, the parameters are updated by

$$\theta_t = \theta_{t-1} - \eta \nabla_\theta \ell_{\text{self}}(z_t; \theta_{t-1}), \tag{2}$$

with learning rate $\eta$. Through sequential updates, $f_{\text{adapt}}$ implicitly retains information from past visual observations, thereby capturing temporal context. In Section 4.6.2, we provide empirical evidence that incremental parameter updates of the value function estimator are more effective than updates performed at the trajectory level.

### 3.1.3 ZERO-SHOT VALUE FUNCTION ESTIMATOR

After test-time adaptation, a meta-learned linear projection $P_Q \in \mathbb{R}^{d' \times 2d}$ maps the input $z_t$ into an adaptation space $\mathbb{R}^{d'}$. Then, this representation is passed through the adaptation module $f_{\text{adapt}}$, followed by a regression head $h$ that outputs predicted values of the goal-conditioned value function:

$$V(z_t; g) = h(f_{\text{adapt}}(P_Q z_t; \theta_t)) \tag{3}$$

The regression head $h$ is a two-layer multilayer perceptron (MLP), following value-function estimator architectures used in deep reinforcement learning (Espeholt et al., 2018). The network is trained using normalized progress labels $y_t$ from expert demonstrations as described in Section 2.1, with a supervised prediction loss $\ell_{\text{pred}}$ defined as the mean squared error between predicted values $V(z_t; g)$ and targets $y_t$. At inference, the regression head $h$ remains frozen, while the adaptation module $f_{\text{adapt}}$ is updated online using the self-supervised objective. This enables the value function estimator to generalize in a zero-shot setting without requiring task-specific demonstrations.

## 3.2 TRAINING PROCEDURE

We train the value function estimator with gradient-based meta-learning, optimizing a self-supervised loss $\ell_{\text{self}}$ such that test-time adaptation improves the supervised prediction loss $\ell_{\text{pred}}$ (Sun et al., 2024). Training is performed on diverse sub-trajectories selected via dissimilarity-based sampling, which mitigates shortcut learning and encourages reliance on semantic and temporal cues.

### 3.2.1 META-LEARNING

During training, the adaptation module $f_{\text{adapt}}$ is updated online at each timestep using the self-supervised loss $\ell_{\text{self}}$. Following the gradient-based meta-learning paradigm (Finn et al., 2017), we differentiate through these adaptation updates with respect to the supervised prediction loss $\ell_{\text{pred}}$. This optimizes the initialization $\theta_0$ of $f_{\text{adapt}}$, the linear projections $P_K, P_V, P_Q$, and the regression head $h$. Formally, the total training loss combines the supervised prediction loss $\ell_{\text{pred}}$ with the self-supervised loss $\ell_{\text{self}}$, weighted by a scalar $\lambda$. The learned parameters $\theta_0$ serve as the initialization for adaptation at inference time. This approach enables the value function estimator to learn to adapt its internal representations at test time to improve value function estimation.

### 3.2.2 DISSIMILARITY-BASED SAMPLING

Expert visual trajectories often contain consecutive frames that are highly redundant, as noted in prior work on video action recognition (Wang et al., 2018). Such redundancy can encourage the value function estimator to exploit shortcut cues (Geirhos et al., 2020; Lee et al., 2021b) by overfitting to frequently occurring late-stage visual patterns. To mitigate this, we propose a dissimilarity-based sampling strategy that constructs mini-batches from the most visually diverse sub-trajectories within each trajectory. This increases intra-batch variance and acts as a form of importance sampling that favors semantic diversity.

Formally, given a sequence of multimodal representations $\{z_1, \ldots, z_T\}$, we extract fixed-length sub-trajectories using a sliding window of size $w_{\text{tr}}$ and stride $s$, yielding a candidate set $\mathcal{W}$. A diverse subset of size $k$ maximizes pairwise dissimilarity:

$$\mathcal{W}' = \underset{\mathcal{U} \subset \mathcal{W}, \, |\mathcal{U}|=k}{\arg\max} \sum_{\{w^i, w^j\} \in \binom{\mathcal{U}}{2}} \|w^i - w^j\|_2^2, \tag{4}$$

where each $w^i$ is a sub-trajectory of length $w_{\text{tr}}$. Solving Eq. 4 is intractable, requiring enumeration of all $\binom{|\mathcal{W}|}{k}$ subsets. To approximate this objective efficiently, we adopt a scoring-based heuristic, assigning each window $w \in \mathcal{W}$ a score equal to its total dissimilarity with all others:

$$s(w) = \sum_{v \in \mathcal{W}} \|w - v\|_2^2, \qquad \mathcal{W}' = \underset{\mathcal{U} \subset \mathcal{W}, \, |\mathcal{U}|=k}{\arg\max} \sum_{w \in \mathcal{U}} s(w). \tag{5}$$

Thus, by computing a diversity score $s$ for each window and selecting the $k$ highest-scoring windows, we obtain a heuristically diverse subset with polynomial-time complexity. The overhead is negligible compared to model training, as shown by our complexity analysis in Appendix G. We empirically validate dissimilarity-based sampling against full-trajectory sampling in the ablation study (Section 4.6.1), showing improved ability to distinguish expert from non-expert visual trajectories.

## 4 EXPERIMENTS

Our evaluation demonstrates the effectiveness of VITA across core capabilities of goal-conditioned value functions: (i) generalization in value estimation for real-world robotic manipulation under distribution shifts in task, environment, and embodiment; (ii) differentiation between expert and non-expert visual trajectories in real-world robotic manipulation tasks; (iii) effective zero-shot reward shaping for offline RL in Meta-World MT10, a simulated benchmark for multi-task robot learning.

### 4.1 TRAINING SETUP

We train VITA on expert visual trajectories paired with natural language task descriptions from the BridgeData V2 dataset (Walke et al., 2023), which spans a wide range of manipulation tasks, environments, and robot embodiments. In particular, we use a curated subset consisting of 2,986 expert demonstrations covering pick-and-place manipulation tasks, but it does not include folding, sweeping, or stacking tasks. All demonstrations are collected using a single robot embodiment (WidowX 250) across 4 configurations of the ToyKitchen environment. An additional 287 expert demonstrations are held out as the in-distribution test set, referred to as tk_pnp. Appendix A includes examples from the training set. We use OpenCLIP ViT-B/32 as the frozen backbone for encoding video frames and task descriptions. The training objective combines $\ell_{\text{pred}}$ with $\ell_{\text{self}}$, weighted by

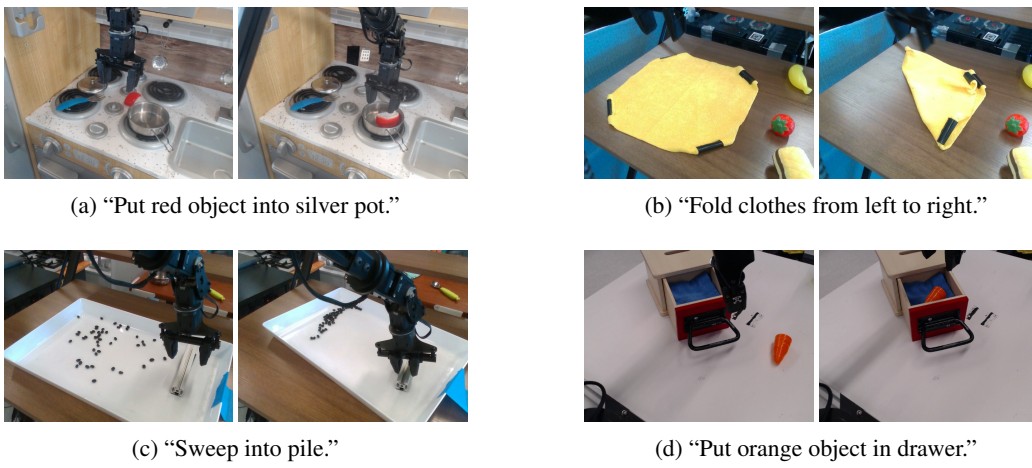

(a) "Put red object into silver pot."

(b) "Fold clothes from left to right."

(c) "Sweep into pile."

(d) "Put orange object in drawer."

Figure 3: Examples of visual trajectories paired with task descriptions under different distribution shifts. (a) In-distribution. (b, c) Environment shift. (d) Embodiment and environment shift.

$\lambda_{\text{self}} = 0.5$. During training, we use dissimilarity-based sampling with window size $w_{\text{tr}} = 8$, number of sub-trajectories $k = 8$, and stride $s = 1$. At test time, we use one gradient step ($t_{\text{ep}} = 1$) using $\lambda_{\text{self}}$ with learning rate $\eta = 0.1$. Further details and hyperparameter sweeps are reported in Appendix D.

## 4.2 BASELINES

We compare VITA against value functions based on contrastive and autoregressive VLMs. VLM-CL computes zero-shot value estimates from cosine similarity between CLIP representations of each frame and task description (Baumli et al., 2023). VLM-RM regularizes CLIP embeddings by projecting features along the direction from a generic reference prompt to the task prompt (Rocamonde et al., 2024). CLIP-FT trains a supervised regression head on frozen CLIP multimodal representations. CLIP-GRU trains a Gated Recurrent Unit (GRU) (Cho et al., 2014) on frozen CLIP multimodal representations. GVL (Ma et al., 2024) is the state-of-the-art zero-shot value function that leverages autoregressive VLMs. In our experiments, GVL-0S refers to the zero-shot setting, where the VLM is prompted with only the test trajectory and task description. GVL-1S corresponds to a one-shot in-context setting, where a full shuffled trajectory from the training distribution, along with its progress labels, is provided as an example. We follow GVL in using Gemini 1.5 Pro (`gemini-1.5 Pro-latest`) (Gemini Team et al., 2024) with its proposed prompt template. We also tested Qwen-VL 2.5 (Yang et al., 2024), which failed to overcome temporal bias, and GPT-4o (Hurst et al., 2024), which refused to produce value estimates. Additional implementation details are provided in Appendix D.

## 4.3 EVALUATING GENERALIZATION UNDER DISTRIBUTION SHIFTS

We evaluate the ability of progress estimators to generalize across novel environments, tasks, and robot embodiments using subsets of BridgeData V2 curated to introduce variation along these axes. Environment shifts involve changes in scene layout or background. For example, `lm_pnp` is a pick-and-place task in front of a laundry machine, while `td_fold`, `ft_fold`, and `rd_fold` feature cloth folding on different surfaces. `ms_sweep` introduces a long-horizon sweeping task in a confined tray. Embodiment shifts are evaluated using the DeepThought robot, which differs from the training embodiment (WidowX 250) in both morphology and camera perspective. `dt_tk_pnp` (pick-and-place) and `dt_tk_stack` (stacking) retain the in-distribution environment with a new embodiment, while `dt_ft_stack` (stacking) and `dt_rd_pnp` (drawer pick-and-place) involve both embodiment and environment shifts. Each dataset includes 200 expert trajectories with task descriptions, except for `ms_sweep`, which contains 100 due to its size limitation. Appendix B provides a full description of our evaluation datasets. We evaluate the performance of a value function estimator using the *Value Order Correlation* (VOC) metric (Ma et al., 2024), which measures the alignment between the predicted progress towards task completion and the chronological order of the frames in a visual

Table 1: VOC scores for value function estimation under distribution shifts. **ID** = In-Distribution, **ES** = Environment Shift, **EM** = Embodiment Shift, **ES & EM** = Both Shifts.

| Shift | Dataset | VLM-CL | VLM-RM | CLIP-FT | GVL-0S | GVL-1S | CLIP-GRU | VITA |
|---|---|---|---|---|---|---|---|---|
| ID | tk_pnp | 0.038 | 0.029 | 0.251 | 0.269 | 0.252 | 0.773 | **0.782** |
| ES | lm_pnp | 0.017 | 0.033 | 0.149 | 0.305 | 0.272 | 0.676 | **0.725** |
| | td_fold | 0.031 | 0.072 | 0.152 | 0.326 | 0.318 | 0.674 | **0.709** |
| | ft_fold | 0.108 | 0.099 | 0.162 | 0.331 | 0.387 | **0.693** | 0.658 |
| | rd_fold | 0.095 | 0.055 | 0.126 | 0.372 | 0.406 | **0.726** | 0.606 |
| | ms_sweep | -0.129 | -0.226 | 0.148 | 0.158 | 0.150 | 0.434 | **0.490** |
| EM | dt_tk_pnp | 0.042 | -0.041 | 0.149 | 0.258 | 0.211 | **0.856** | 0.820 |
| | dt_tk_stack | 0.035 | 0.046 | 0.099 | 0.254 | 0.277 | 0.667 | **0.708** |
| ES & EM | dt_ft_stack | 0.026 | 0.028 | 0.049 | 0.212 | 0.249 | 0.674 | **0.698** |
| | dt_rd_pnp | 0.023 | 0.041 | 0.211 | 0.329 | 0.316 | **0.747** | 0.695 |

trajectory. Formally, let $p_1, \ldots, p_T$ denote the predicted progress values for a trajectory of $T$ frames, and let $r_t = t$ represent the temporal index. VOC is defined as the Spearman rank correlation $\rho_s$ between the predicted values and the temporal indices.

In Table 1, the results show that both VITA and CLIP-GRU consistently outperform GVL-0S and GVL-1S, highlighting the limitations of pre-trained autoregressive VLMs and the importance of preserving temporal order for effective progress estimation. While both demonstrate effective progress estimation, VITA outperforms CLIP-GRU on the majority of evaluation datasets (6 out of 10). Both VITA and CLIP-GRU outperform CLIP-based baselines that lack temporal modeling (CLIP-FT, VLM-CL, VLM-RM), indicating the impact of encoding temporal history on progress estimation. CLIP-FT performs comparably to GVL on the in-distribution task but fails to generalize to out-of-distribution settings. Both VLM-CL and VLM-RM perform poorly, likely due to their lack of temporal modeling. For GVL-0S and GVL-1S, in some cases, in-context examples improve performance, while in others, they provide no benefit or even degrade performance. Both GVL methods perform well on folding tasks (td_fold, ft_fold, rd_fold), but fail to achieve comparable performance on stacking (dt_tk_stack, dt_ft_stack) and pick-and-place (dt_tk_pnp, lm_pnp) tasks, suggesting that the autoregressive VLM used in GVL may be biased toward folding-like robotic manipulations. In contrast, VITA achieves consistent performance across all task types and distribution shifts, indicating stronger generalization than in-context learning.

All methods, except CLIP-FT, perform worse on the long-horizon task ms_sweep, but VITA achieves the highest VOC score. In particular, VITA outperforms CLIP-GRU, indicating that encoding temporal history via test-time adaptation is more effective than modeling it through recurrent hidden states for long-horizon robotic manipulation tasks. Under embodiment shifts, VITA demonstrates strong generalization. In dt_tk_pnp, which uses a different robot embodiment but shares the same environment and tasks as the training set, both VITA and CLIP-GRU exceed their in-distribution performance, indicating that learning temporal context can effectively transfer across robot embodiments. In contrast, GVL-1S performs poorly on both dt_tk_pnp and dt_tk_stack, suggesting that few-shot learning fails to generalize under embodiment shifts in our evaluation setup.

## 4.4 DIFFERENTIATING EXPERT FROM NON-EXPERT TRAJECTORIES

Beyond generalization, we evaluate VITA's robustness by testing its ability to distinguish expert from suboptimal visual trajectories, assigning lower progress scores to the latter. To this end, we compare model predictions on expert and scripted (non-expert) visual trajectories collected from the same in-distribution setting of BridgeData V2. The scripted trajectories are generated in the ToyKitchen environment using a heavily randomized controller (Walke et al., 2023), under the same embodiment and environmental configuration as the expert demonstrations. These include pick-and-place tasks involving object categories that may overlap with the training set, such as general objects (sc_pnp_obj), rigid items (sc_pnp_robj), soft toys (sc_pnp_stoy), utensils (sc_pnp_uten), and a cloth (sc_pnp_cloth). While these trajectories match the training distribution in task, embodiment, and environment, their behavior deviates from the smooth and monotonic progress

| Method | BinVOC |
|--------|--------|
| VLM-CL | 0.40 |
| VLM-RM | 0.00 |
| CLIP-FT | 0.80 |
| GVL-0S | **1.00** |
| GVL-1S | **1.00** |
| CLIP-GRU | 0.80 |
| VITA | **1.00** |

(a) Expert vs. Non-Expert (BinVOC).

| Method | IQM | 95% CI |
|--------|-----|--------|
| VLM-CL | 0.760 | [0.722, 0.791] |
| VLM-RM | 0.746 | [0.718, 0.771] |
| CLIP-FT | 0.785 | [0.759, 0.809] |
| CLIP-GRU | 0.777 | [0.734, 0.814] |
| META-WL | 0.779 | [0.750, 0.804] |
| VITA | **0.815** | [0.785, 0.838] |

(b) Offline RL on MT10 (IQM).

Table 2: (a) Success in distinguishing expert from scripted robot demonstrations, measured by average BinVOC across 5 in-distribution scripted datasets. (b) Offline RL performance on the Meta-World MT10 benchmark measured by interquartile mean (IQM) with 95% stratified bootstrap CIs.

typically exhibited by expert demonstrations. Appendix C includes examples from the scripted dataset. As scripted datasets are generated by a randomized controller, they are not designed to reflect quantifiable levels of suboptimality. Nonetheless, an effective value function estimator should assign lower VOC scores to suboptimal trajectories relative to expert ones when evaluated in the same setting. We measure discriminative success with *BinVOC*, a binary evaluation metric defined as $\mathbf{1}\left[\text{VOC}_{\text{exp}} > \text{VOC}_{\text{subopt}}\right]$, which is 1 when expert trajectories achieve higher VOC than suboptimal ones and 0 otherwise.

The model's discriminative performance is evaluated using BINVOC, averaged over all scripted datasets. Performance on expert visual trajectories ($\text{VOC}_{\text{exp}}$) is measured on the in-distribution test set tk_pnp, which features pick-and-place tasks similar to those in the scripted evaluation. VITA, GVL-0S, and GVL-1S achieve perfect discrimination, consistently assigning higher VOC scores to expert trajectories across all tasks. VITA outperforms CLIP-GRU, indicating that encoding temporal memory in recurrent hidden states may be more prone to overfitting to temporal shortcuts compared to implicit memory via sequential test-time updates. CLIP-based baselines (VLM-CL, VLM-RM) likely underperform due to the absence of temporal modeling. CLIP-FT performs more reliably, but fails on one task, suggesting sensitivity to object distribution shifts.

## 4.5 ZERO-SHOT REWARD SHAPING FOR OFFLINE RL

We evaluate our value function estimator on the Meta-World MT10 benchmark (Yu et al., 2020), which consists of ten diverse robotic manipulation tasks, including pick-and-place, door opening, and pushing. This experiment is designed to evaluate VITA's zero-shot reward shaping for offline RL. VITA is trained on real-world robotic data and evaluated zero-shot in this simulated setting. For each task in Meta-World MT10 benchmark, we generate 20 expert visual demonstrations using Meta-World's expert policies. The goal-conditioned value estimator is then used to define a dense reward for each visual observation. Policies are trained with Implicit Q-Learning (IQL) (Kostrikov et al., 2021) in the offline RL setting. We repeat expert data generation and policy training across 10 random seeds. After training, each multi-task policy is evaluated on 20 rollouts per task, and we compute the average return across episodes for each (task, seed) pair. We report the interquartile mean (IQM) across all pairs, together with 95% stratified bootstrap confidence intervals, following the evaluation protocol proposed by Agarwal et al. (2021). The training details and evaluation protocol are further described in Appendix E.

Table 2 reports performance on the MT10 benchmark for the baselines under the evaluation protocol described above. VITA achieves the highest IQM return (0.815), outperforming all CLIP-based baselines. A direct comparison with GVL is infeasible at scale, since it relies on Gemini-1.5, a proprietary autoregressive VLM with prohibitively high inference cost in RL settings, while open-source alternatives did not yield reliable progress estimates in our preliminary tests. We also report the fuzzy-logic dense reward provided by Meta-World (META-WL), which achieves 0.779. Despite its strong performance, META-WL is outperformed by VITA, indicating that a value estimator trained on real-world data can generalize effectively to simulated reward shaping.

## 4.6 ABLATION STUDIES

### 4.6.1 EFFECT OF DISSIMILARITY-BASED SAMPLING ON DISCRIMINATIVE PERFORMANCE

We empirically validate the effectiveness of the proposed dissimilarity-based sampling by comparing the impact of different sub-trajectory sampling strategies on VITA's ability to distinguish expert from non-expert demonstrations. In particular, full-trajectory sampling (Sun et al., 2024) computes the adaptation loss over all overlapping sub-trajectories with stride $s = 1$, which in our setup leads to overfitting to global temporal shortcuts. Random sampling improves diversity by selecting sub-trajectories uniformly, but lacks semantic diversity. In contrast, dissimilarity-based sampling explicitly promotes semantic diversity, resulting in better generalization. Our proposed method, which selects sub-trajectories that maximize pairwise dissimilarity, outperforms both full-trajectory and random sampling in differentiating expert and non-expert visual trajectories. For further details, please refer to Appendix F.1.

### 4.6.2 EFFECT OF TEMPORAL MEMORY ON VALUE FUNCTION ESTIMATION

We analyze how different test-time adaptation strategies for incorporating temporal context impact value function estimation. VITA performs step-by-step adaptation without reset (implicit memory), incrementally updating parameters from $\theta_{t-1}$. This allows the value function estimator to encode trajectory history directly into its parameters, capturing both temporal order and accumulated temporal context. We compare VITA with three alternatives: (i) *trajectory-level adaptation (TTT-TR)*, which applies a single gradient update computed over the full trajectory. This produces a batched update that averages the loss across all frames, thereby discarding temporal order; (ii) *memoryless adaptation (TTT-RS)*, which resets the adaptation module to $\theta_0$ at every step. By adapting only to the current visual observation, this approach prevents any temporal information from being carried forward; (iii) *explicit memory (TTT-EX)*, which also resets the adaptation module at every step but updates using a local window of recent frames matching the window size used during meta-training ($w_{tr} = 8$). Across all datasets, VITA consistently outperforms trajectory-level, memoryless, and explicit-memory methods, demonstrating the effectiveness of incrementally accumulating temporal context for value function estimation. We provide the full analysis in Appendix F.2.

## 5 RELATED WORK

### 5.1 VLMS AS VALUE FUNCTION ESTIMATORS

Contrastive VLMs have been applied as zero-shot reward models (Baumli et al., 2023; Rocamonde et al., 2024) and goal-conditioned value functions (Ma et al., 2023b; 2024), using frame-level similarity scores but lacking temporal modeling. Some works fine-tune CLIP on domain-specific video datasets (Fan et al., 2022; Jiang et al., 2024) to generate dense rewards, requiring supervision that limits their zero-shot applicability. RoboCLIP (Sontakke et al., 2023) enables one-shot reward specification without fine-tuning, but yields sparse rewards and still requires demonstrations. Large-scale multimodal pre-training approaches (Alakuijala et al., 2024; Li et al., 2024; Ma et al., 2023b) learn language-conditioned value functions by relying on scale for generalization. ReWiND (Zhang et al., 2025b) trains a cross-modal sequential transformer on augmented large-scale data with video rewinding, combined with a few in-domain demonstrations to improve generalization. In contrast, VITA adapts a pre-trained contrastive vision–language encoder at inference, capturing both temporal and semantic context through test-time adaptation, without requiring large-scale multimodal pre-training or additional domain-specific demonstrations for generalization. VITA could be applied to any pre-trained vision–language representation that encodes a universal value function to enhance its generalization and temporal reasoning via meta-learned test-time adaptation. Autoregressive VLMs (Alayrac et al., 2022) leverage in-context learning over trajectories and have been applied to success detection (Du et al., 2023) and goal-conditioned value functions (Ma et al., 2024). GVL (Ma et al., 2024) mitigates monotonic bias from ordered pre-training data by shuffling frames at inference, but this discards temporal order, which is essential for temporal reasoning and for distinguishing visually similar yet temporally distinct states. Our approach instead preserves chronological order while capturing temporal context through test-time adaptation, avoiding such shortcuts.

## 5.2 TEST-TIME ADAPTATION

Test-time adaptation methods provide parameter-efficient solutions for adapting VLMs without the need for full fine-tuning. Prompt-based methods optimize model parameters at test time based on the task context, although their underlying mechanisms are opaque (Zhou et al., 2022; Khattak et al., 2023; Ma et al., 2023a). Lim et al. (2025) suggested that CLIP representations could support test-time adaptation, based on the observation that CLIP encodes human-interpretable concepts discovered via mechanistic interpretability (Bricken et al., 2023). CLIP-based similarity has also been used as a reward signal, with parameters optimized at test time via reinforcement learning (Zhao et al., 2024). Another approach is test-time training (Sun et al., 2020), where an adaptation module is updated at each timestep via a self-supervised loss. Unlike meta-reinforcement learning methods (Finn et al., 2017; Bauer et al., 2023), which require task-specific adaptation episodes during training, test-time training enables direct online adaptation during inference, without the need for task labels.

## 6 DISCUSSION, LIMITATIONS, AND FUTURE WORK

### 6.1 DISCUSSION

We show that test-time adaptation enables value functions for robotic manipulation to generalize across distribution shifts in task, environment, and embodiment. In addition, VITA can distinguish expert from non-expert visual trajectories and perform zero-shot reward shaping, enabling downstream applications in RL and imitation learning. By updating sequentially over a trajectory, VITA encodes history into its parameters, capturing temporal and semantic context more effectively than CLIP-based baselines and in-context learning with autoregressive VLMs. VITA outperforms CLIP-GRU on expert–non-expert discrimination and offline RL, indicating that implicit memory via sequential test-time updates generalizes more effectively than explicitly encoding temporal history in recurrent hidden states for zero-shot value function estimation. Pre-trained autoregressive VLMs (GVL), despite their generalization capabilities, are not explicitly trained for progress estimation, which requires temporal reasoning over visual trajectories. In our setup, the inference-time overhead of per-timestep adaptation is negligible, as only a lightweight adaptation module is updated at test time, which does not affect real-time applicability.

### 6.2 LIMITATIONS AND FUTURE WORK

Test-time adaptation improves generalization of zero-shot progress estimation across unseen tasks and environments in our experiments, but it can still face challenges in settings with high execution variability or extended durations. Although the adaptation overhead is negligible due to the lightweight adaptation module, updating a value function estimator at every timestep may be potentially unsafe during deployment, limiting applicability in real-time scenarios. In future work, we plan to explore alternative approaches for test-time adaptation of vision–language models, particularly in the context of training agents within world models. In addition, applying VITA to real-time closed-loop control settings and complex RL environments remains a promising direction for future work. VITA could enhance generalization and temporal reasoning for any domain-specific pre-trained multimodal representation via meta-learned test-time adaptation, as it is agnostic to the underlying vision–language encoder. While we provide empirical evidence that dissimilarity-based sampling mitigates shortcut learning, a theoretical analysis of how diversity-based sampling influences shortcut learning remains a promising direction. This lies beyond the scope of our core contribution, which introduces a value function estimator that meta-learns a test-time adaptation mechanism to improve zero-shot performance and temporal reasoning.

## ACKNOWLEDGMENTS

We thank Daniel Furelos-Blanco, Roko Parac, and Frederik Kelbel for their helpful comments on an earlier draft. We also thank Vassilis Digalakis for valuable discussions. This work was supported by UKRI (EP/Y037111/1) as part of the ProSafe project (EU Horizon 2020, MSCA, grant no. 101119358).

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

## A    TRAINING DATASET

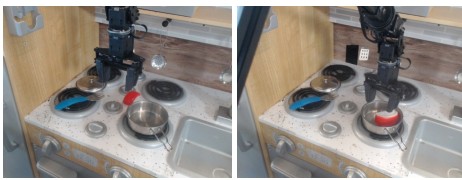

(a) "Put red object into silver pot."

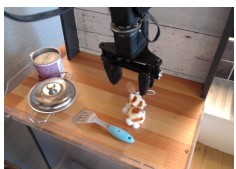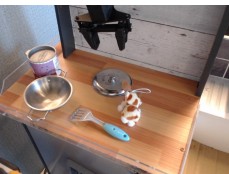

(b) "Take silver bowl lid from table."

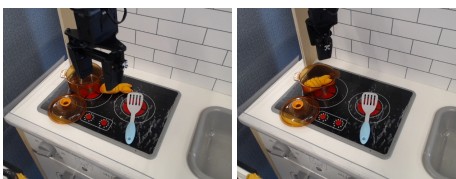

(c) "Place yellow object inside pot."

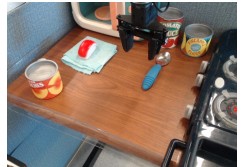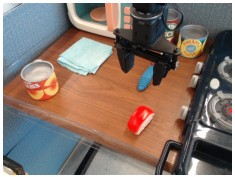

(d) "Move red object from pot to left burner."

Figure 4: Each subfigure shows the start and end frames from an expert demonstration used for training, along with its natural language task description. Demonstrations are collected across four distinct ToyKitchen environments.

## B    EVALUATION DATASETS

Table 3: Dataset descriptions with task type, environment, and embodiment.

| Dataset | Task Type | Environment | Embodiment |
|---|---|---|---|
| tk_pnp | pick-and-place | toy kitchen | WidowX 250 |
| lm_pnp | pick-and-place | laundry machine | WidowX 250 |
| td_fold | fold cloth | tabletop (dark wood) | WidowX 250 |
| ft_fold | fold cloth | folding table | WidowX 250 |
| rd_fold | fold cloth | robot desk | WidowX 250 |
| ms_sweep | sweep | folding table (tray) | WidowX 250 |
| dt_tk_pnp | pick-and-place | toy kitchen | DeepThought |
| dt_tk_stack | stack blocks | toy kitchen | DeepThought |
| dt_ft_stack | stack blocks | folding table | DeepThought |
| dt_rd_pnp | pick-and-place | robot desk (drawer) | DeepThought |

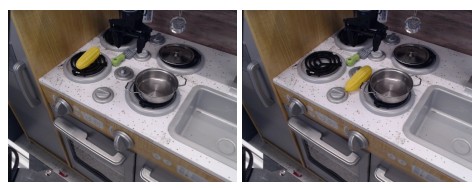

(a) `dt_tk_ms`: "Move yellow object from burner to center."

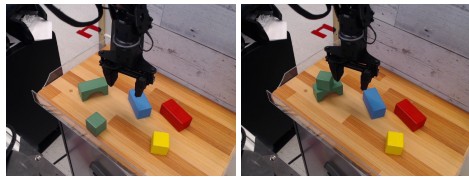

(b) `dt_tk_stack`: "Place green cube on top of arch."

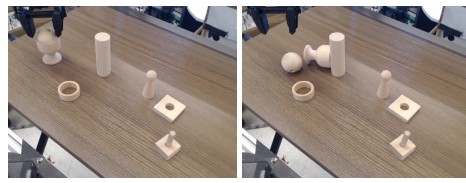

(c) `dt_ft_stack`: "Move egg to table."

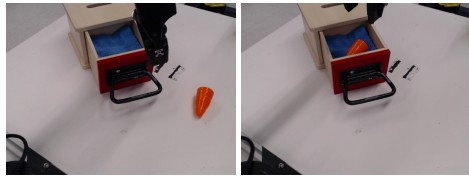

(d) `dt_rd_pnp`: "Put orange object in drawer."

Figure 5: Each subfigure shows the start and end frames from an evaluation trajectory under embodiment shift, along with its natural language task description. The top row depicts tasks in the same environment (ToyKitchen) using a different robot (DeepThought), while the bottom row includes tasks that also involve new environments.

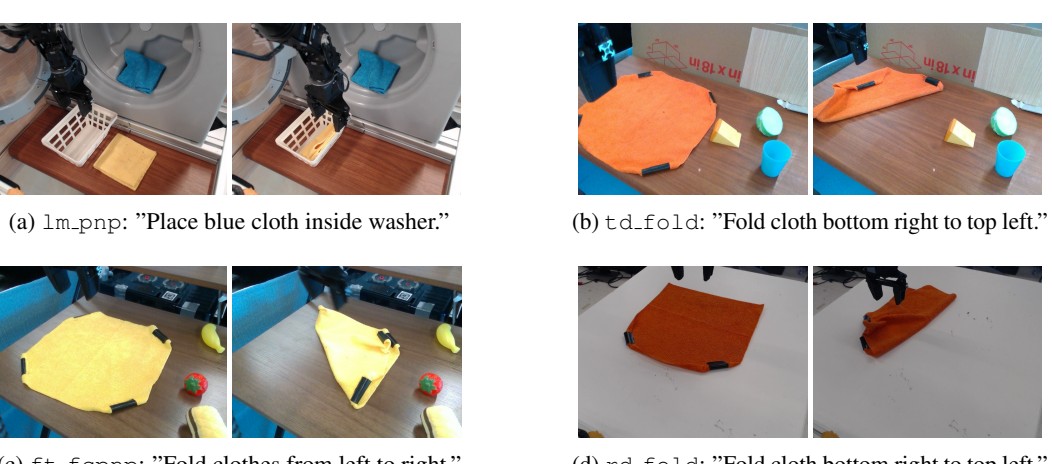

(a) `lm_pnp`: "Place blue cloth inside washer."

(b) `td_fold`: "Fold cloth bottom right to top left."

(c) `ft_fcpnp`: "Fold clothes from left to right."

(d) `rd_fold`: "Fold cloth bottom right to top left."

(e) `ms_ft_sweep`: "Sweep into pile."

Figure 6: Each subfigure shows the start and end frames from an evaluation trajectory under environment shift, along with its natural language task description. All tasks are performed using the same robot embodiment across visually and structurally different environments.

## C   SCRIPTED DATASET

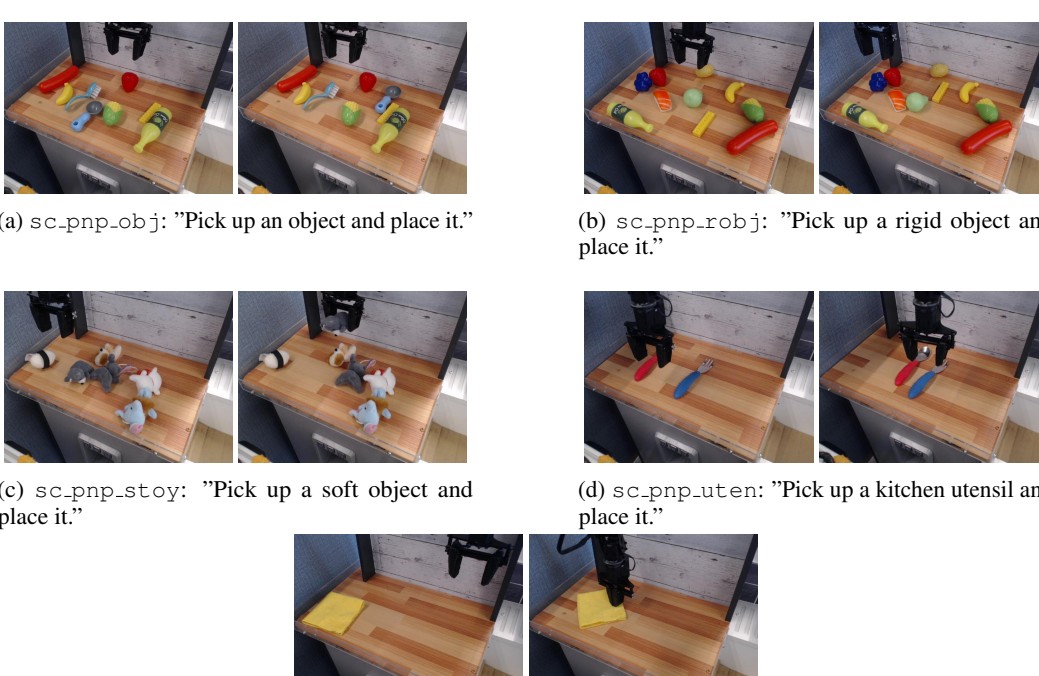

(a) sc_pnp_obj: "Pick up an object and place it."

(b) sc_pnp_robj: "Pick up a rigid object and place it."

(c) sc_pnp_stoy: "Pick up a soft object and place it."

(d) sc_pnp_uten: "Pick up a kitchen utensil and place it."

(e) sc_pnp_cloth: "Move the cloth."

Figure 7: Each subfigure shows the start and end frames from a scripted evaluation trajectory collected in the ToyKitchen environment using the WidowX 250 robot. All tasks involve pick-and-place behavior with varying object categories. We use generic task descriptions due to the lack of publicly available annotations.

## D   MODEL ARCHITECTURE

### D.1   MODEL AND TRAINING DETAILS

We use the OpenCLIP ViT-B/32 encoder pre-trained on the OpenAI dataset as a frozen backbone. Each visual observation and task description is encoded using the CLIP vision and text encoders to produce their representation, respectively. We opted for a joint (concatenation-based) representation over an element-wise product based on improved validation performance. The adaptation module $f_{\text{adapt}}$ is a two-layer residual MLP with GELU activation and a projection dimension $d' = 64$. The model is trained using the AdamW optimizer with a learning rate of $1 \times 10^{-4}$, weight decay of $1 \times 10^{-4}$, and a cosine learning rate schedule with 10% warmup. We use a batch size of 32 and pad all trajectories to a maximum length of 120 frames (matching the longest sequence in the training set). The weighting coefficient $\lambda_{\text{self}}$ of the self-supervised loss in the total training objective is set to 0.5, selected based on validation performance. We train for 5 epochs, as validation VOC typically plateaus early, with extended training providing no further improvement. For dissimilarity-based sampling, we set the stride to $s = 1$, the sub-trajectory length to $w_{\text{tr}} = 8$, and the number of selected windows to $k = 8$, unless otherwise specified. All experiments were run on NVIDIA RTX 6000 Ada Generation GPUs.

### D.2   TEST-TIME TRAINING HYPERPARAMETERS

At inference time, we adapt only the temporal adaptation module $f_{\text{adapt}}$, using the same projection dimension ($d' = 64$) as in training. We perform a single gradient step ($t_{\text{ep}} = 1$) with

Table 4: Effect of test-time learning rate $\eta$ and number of adaptation steps $t_{\text{ep}}$ on VITA performance on the validation set measured by VOC.

| $\eta$ | $t_{\text{ep}} = 1$ | $t_{\text{ep}} = 5$ | $t_{\text{ep}} = 10$ |
|---|---|---|---|
| 0.001 | 0.225 | 0.345 | 0.453 |
| 0.01 | 0.454 | **0.744** | **0.781** |
| 0.05 | 0.746 | 0.670 | 0.515 |
| 0.1 | **0.783** | 0.512 | 0.369 |
| 1.0 | 0.299 | 0.183 | 0.181 |
| 5.0 | 0.169 | 0.182 | 0.157 |

a learning rate $\eta = 0.1$. This configuration was selected from a sweep over learning rates $\{5.0, 1.0, 0.1, 0.05, 0.01, 0.001\}$ and adaptation steps $\{1, 5, 10\}$, based on validation performance reported in Table 4. The results show that both (i) a single update with a larger learning rate ($\eta = 0.1$, $t_{\text{ep}} = 1$) and (ii) multiple updates with a smaller learning rate ($\eta = 0.01$, $t_{\text{ep}} = 10$) achieve comparable validation performance under test-time training. We select the former because it requires fewer adaptation steps while achieving slightly better results.

### D.3 BASELINES

VITA uses $t_{\text{ep}} = 1$, projection dimension $d' = 64$, and learning rates $\eta$ of 0.1, selected via hyperparameter tuning as described previously. CLIP-FT follows VITA's architecture, but excludes the adaptation module and self-supervised loss. In particular, CLIP-FT uses a frozen CLIP encoder, followed by a linear projection and a two-layer MLP to predict task progress. The model is trained with standard supervised regression, without meta-learning or test-time adaptation. To increase expressivity, it uses an $8\times$ larger projection matrix and $10\times$ more training steps than our method (Lin et al., 2022). For VLM-RM, we use a baseline prompt that describes the environment. CLIP-GRU follows VITA's architecture (frozen CLIP encoder and regression head $h$), but replaces the test-time adaptation module $f_{\text{adapt}}$ with a Gated Recurrent Unit (GRU) module that explicitly encodes temporal history through recurrent hidden states. The GRU module has a single layer with hidden size 64, and its final hidden state is passed to the regression head $h$. For both GVL-0S and GVL-1S, we use the latest version of Gemini 1.5 Pro, `gemini-1.5-pro-latest` (Gemini Team et al., 2024), whereas the original GVL implementation used an earlier release, `gemini-1.5-pro`. We also evaluated GVL using the open-source autoregressive VLM `Qwen-VL 2.5` (Yang et al., 2024), which failed to overcome temporal bias despite frame shuffling, while GPT-4o (Hurst et al., 2024) consistently declined to produce scalar progress estimates in our setup. All evaluations followed the prompt template introduced in the original GVL work (Ma et al., 2024).

## E OFFLINE RL SETUP

### E.1 TRAINING CONFIGURATION

We use Implicit Q-Learning (IQL) with the following settings, kept fixed across all tasks. Expectile regression is set to $\tau = 0.7$, with advantage weighting temperature 3.0 and clipping threshold 100.0. The actor and critic use learning rates $1 \times 10^{-4}$ and $3 \times 10^{-4}$, respectively, with batch size 256. Policies are trained for 100,000 gradient steps, with evaluation every 10,000 steps on 20 rollouts per task (horizon 150). Implementation follows `d3rlpy`, with configuration: `expectile=0.7, weight_temp=3.0, max_weight=100.0, actor_lr=1e-4, critic_lr=3e-4, batch_size=256`.

### E.2 EVALUATION PROTOCOL

Following Agarwal et al. (Agarwal et al., 2021), we report the interquartile mean (IQM) of returns across all task–seed pairs. The IQM computes the mean of the middle 50% of scores, which reduces sensitivity to outliers compared to the mean or median. For statistical robustness, we estimate

95% confidence intervals using stratified bootstrap with 10,000 resamples, stratified by task. This procedure is applied consistently across all methods in Meta-World MT10 experiments.

## F  ABLATION STUDIES

### F.1  EFFECT OF DISSIMILARITY-BASED SAMPLING ON DISCRIMINATIVE PERFORMANCE

Table 5: Disc@VOC results across scripted datasets under different training-time sampling strategies for test-time adaptation. A checkmark indicates successful discrimination between expert and suboptimal trajectories. Sampling methods **FT** (full-trajectory), **Rand** (random sampling), and **Diss** (dissimilarity-based sampling) are parameterized by $(w_{\text{tr}}, k)$.

| Dataset | FT (8,–) | Rand (8,8) | Diss (8,4) | Diss (4,8) | Diss (8,8) |
|---|---|---|---|---|---|
| sc_pnp_obj | | ✓ | ✓ | | ✓ |
| sc_pnp_robj | ✓ | ✓ | ✓ | ✓ | ✓ |
| sc_pnp_stoy | | | ✓ | | ✓ |
| sc_pnp_uten | ✓ | ✓ | ✓ | ✓ | ✓ |
| sc_pnp_cloth | | ✓ | ✓ | ✓ | ✓ |
| **Avg. Disc@VOC** | 0.20 | 0.8 | **1.0** | 0.60 | **1.0** |

We empirically validate the effectiveness of the proposed dissimilarity-based sampling by comparing the impact of different sub-trajectory sampling strategies. In particular, we evaluate how different sub-trajectory sampling strategies during training impact the ability of our model to discriminate expert from suboptimal trajectories. In Table 5, we compute Disc@VOC for 5 scripted datasets, capturing whether a given model assigns a higher average VOC to expert trajectories (tk_pnp) than to each evaluation dataset. We compare three strategies: (1) full-trajectory sampling applies adaptation over all overlapping sub-trajectories of length $w_{\text{tr}}$ using stride $s$, following prior work (Sun et al., 2024); (2) random sampling uniformly samples $k$ sub-trajectories of length $w_{\text{tr}}$ per video; and (3) our proposed dissimilarity-based sampling constructs a candidate set of sub-trajectories of length $w_{\text{tr}}$ using stride $s$, and selects $k$ sub-trajectories that maximize pairwise dissimilarity in the representation space.

Our results show that dissimilarity-based sampling with $w_{\text{tr}} = 8$ and $k = 8$ yields perfect discriminative performance. Sampling fewer windows ($k = 4$) of size $w_{\text{tr}} = 8$ leads to the same performance, while reducing sub-trajectory length to $w_{\text{tr}} = 4$ reduces effectiveness. The full-trajectory baseline performs the worst, confirming that adapting over all consecutive windows can lead to overfitting to global temporal cues. Random sampling outperforms full-trajectory sampling but remains less reliable than dissimilarity-based selection. These findings highlight the importance of both sub-trajectory diversity and semantic locality in enabling discriminative test-time adaptation.

### F.2  EFFECT OF TEMPORAL MEMORY ON VALUE FUNCTION ESTIMATION

We evaluate four adaptation strategies: (1) TTT-TR, which performs a single gradient update computed over the full trajectory (offline). TTT-TR performs a batched update that averages the loss across all frames, thereby discarding temporal order; (2) TTT-RS, which resets the adaptation module at every step and updates using only the current visual observation (memoryless). TTT-RS prevents any temporal information from being carried forward; (3) TTT-EX, which resets the adaptation module at every step and updates using a local window of recent frames matching the window size used during meta-training ($w_{\text{tr}} = 8$). TTT-EX averages gradients over this window, discarding the temporal ordering within it; (4) VITA (OURS), which updates the adaptation module incrementally at every timestep (implicit memory). VITA forms an implicit memory through the sequential accumulation of temporal history in the adapter parameters. We performed the same test-time hyperparameter sweeps for all variants, as described in Appendix D. We found that VITA, TTT-RS, and TTT-TR achieve optimal performance with a learning rate of 0.1 and a single adaptation step, whereas TTT-EX performs best with a learning rate of 1.0 and one adaptation step.

Table 6: Validation VOC scores for progress estimation under distribution shifts. ID = In-Distribution, ES = Environment Shift, EM = Embodiment Shift, ES+EM = Environment and Embodiment Shift.

| Shift | Dataset | TTT-TR | TTT-RS | TTT-EX | VITA |
|-------|---------|--------|--------|--------|------|
| ID | tk_pnp | 0.1917 | 0.1918 | 0.1957 | **0.7822** |
| ES | lm_pnp | 0.1843 | 0.1854 | 0.1826 | **0.7246** |
| | td_fold | 0.1402 | 0.1393 | 0.1322 | **0.7085** |
| | ft_fold | 0.1302 | 0.1311 | 0.1683 | **0.6583** |
| | rd_fold | 0.1161 | 0.1172 | 0.1154 | **0.6056** |
| | ms_sweep | -0.0225 | -0.0191 | 0.0102 | **0.4898** |
| EM | dt_tk_pnp | 0.2123 | 0.2118 | 0.2069 | **0.8203** |
| | dt_tk_stack | 0.0833 | 0.0830 | 0.0747 | **0.7081** |
| ES+EM | dt_ft_stack | 0.0558 | 0.0537 | 0.0439 | **0.6979** |
| | dt_rd_pnp | 0.1665 | 0.1660 | 0.1559 | **0.6951** |

The results in Table 6 show that TTT-TR, TTT-RS, and TTT-EX yield comparable performance. This indicates that performing batched updates—whether over a short explicit window (TTT-EX) or over the full trajectory (TTT-TR)—performs similarly to the memoryless baseline (TTT-RS). In contrast, VITA consistently outperforms all three variants, underscoring the importance of sequential, per-timestep updates for effective value function estimation in real-world robotic manipulation tasks. In implicit memory (VITA), updating over a local window of frames would cause the most recent observations in that window to be incorporated twice—explicitly through the window and implicitly through the accumulated parameters. In addition, updating less frequently (non-sequentially) would cause information loss because intermediate frames are omitted. Therefore, the implicit-memory test-time training paradigm requires sequential, per-timestep updates without reset.

## G    COMPLEXITY ANALYSIS OF DISSIMILARITY-BASED SAMPLING

Let $\mathcal{W}$ denote the candidate set of size $N_c = |\mathcal{W}|$. To approximate diverse subset selection without the exponential cost of enumerating all $\binom{N_c}{b}$ subsets as described in Eq. 4, we adopt a scoring-based heuristic that assigns each candidate window a diversity score equal to the sum of its pairwise distances to all others (i.e., the row sum of the distance matrix). Let the number of candidate windows be $N_c = T - w_{\text{tr}} + 1$, each represented by a flattened feature vector of dimension $m = w_{\text{tr}} \cdot d$, where $d$ is the dimension of each multimodal representation $z_t$. We compute the full pairwise distance matrix $D \in \mathbb{R}^{N_c \times N_c}$ on the GPU using batched `cdist`, which requires $\mathcal{O}(N_c^2 \cdot w_{\text{tr}} \cdot d)$ operations.

In our experiments, training trajectories are short (mean length 69, max length 120). With a batch size of 32, feature dimension $d = 1024$ (CLIP-based multimodal representation), and window length $w_{\text{tr}} = 8$, the worst-case computational cost is:

$$32 \cdot N_c^2 \cdot w_{\text{tr}} \cdot d \;=\; 32 \cdot 114^2 \cdot 8 \cdot 1024 \;\approx\; 340 \text{ MFLOPs}.$$

This overhead is negligible compared to the cost of a single forward and backward pass through our model, while promoting the selection of diverse windows.

