# OpenReview forum: "VITA: Zero-Shot Value Functions via Test-Time Adaptation of Vision–Language Models"
_ICLR.cc/2026/Conference — ICLR 2026 Poster_

### Official Review · Reviewer_ECyi · 2025-10-30

**Soundness:** 3
**Presentation:** 3
**Contribution:** 3
**Rating:** 6
**Confidence:** 3

**Summary:**

This work proposes using Test-time adaptation modules to improve language-goal conditioned value estimation. Experiments demonstrate that it performs better than GVL (a prompting method) and other VLM based CLIP reward functions.

**Strengths:**

* the paper does a good job explaining TTT
* the idea is rather intuitive -- by gaining more context about the current setting we can produce a better value estimate.
* The paper evaluates both VOC and also running offline RL using the value estimates as reward.
* the data weighting scheme for diverse sampling makes sense.
* the paper performs well on the bridge data setting in comparison to GVL.

**Weaknesses:**

* The paper could benefit substantially from a more detailed figure showing the overall architecture (concatenating clip latents), then how each of the projection matrices are used for TTT, and then how the final observation in the window is used for estimating the value function. I think this would help clarify the total overall flow.
* The paper does not consider other value learning baselines like LIV (ma et al.) which consider langauge and goal representations are also based on CLIP. Generally the paper seems to lack a number of baselines which actually train on the target data like the proposed TTT method. In this regard, LIV seems like a rather natural baseline.
* MetaWorld experiments where the scripted expert demonstrations are used are generally weak as most tasks in the benchmark can be solved by BC on ~10-20 scripted demos! So I'm not sure how significant the policy learning results are (it might be good to include a BC baseline, since IQL is basically weighting the data by the value fn).

**Questions:**

* Just to clarify, TTT is run on each latent observation (concat of visual and lang embed from clip) within a window, and the value function is predicted after the final observation post adaptation.

---

> ### Author Response · Authors · 2025-11-23
> **Response to Reviewer ECyi (Part 1)**
>
> We thank reviewer ECyi for the valuable feedback, the positive review, and for recognizing both the intuitiveness of our work and the strength of our experiments. Our main contribution is a test-time adaptation method that improves both generalization and temporal reasoning of VLM models used for zero-shot goal-conditioned value function estimation, addressing key limitations of current VLM-based value estimators. We address each weakness (W) and question (Q) below. Due to character limits, we have split our response into two separate comments. In this first comment (Part 1), we address the Weaknesses (W1–W3). Please see the subsequent comment (Part 2) for our response to Question Q1.
>
> **W1** :  No detailed architecture figure for TTT.
>
> Thank you for your valuable suggestion. We added a diagram (Figure 2) that visualizes test-time adaptation during inference in detail, including: frozen multimodal representation, meta-learned self-supervised loss parameterized by linear projections, and test-time adaptation updates. In addition, we updated the overview diagram (Figure 1) to present meta-learning in more detail. Both diagrams are included in the revised version. Thank you for helping us improve the clarity of our work.
>
>
> **W2**: Lack of value-learning baselines trained on the target data, such as LIV[1].
>
>
> We thank the reviewer for this suggestion. We implemented an additional baseline (CLIP-GRU) to incorporate their feedback. CLIP-GRU follows VITA’s architecture (frozen CLIP encoder and regression head h), but replaces the test-time adaptation module $f_{\text{adapt}}$ with a Gated Recurrent Unit (GRU) that encodes temporal history through recurrent hidden states. This serves as the most direct "trained baseline" because it uses the same architecture and training data as VITA, differing only in the temporal modeling mechanism. Standard sequence architectures encode temporal history in hidden states (CLIP-GRU) or cached key–value pairs (GVL) while keeping model parameters fixed during inference. In contrast, VITA encodes history implicitly by sequentially updating its parameters at test time.
>
>
> VITA outperforms CLIP-GRU on progress estimation for the majority of evaluation datasets (6 out of 10), particularly on the long-horizon task ms_sweep. Moreover, VITA is more effective at distinguishing expert from non-expert demonstrations (BinVOC: VITA 1.00 vs. CLIP-GRU 0.80). In the Meta-World MT10 benchmark, VITA also outperforms CLIP-GRU for zero-shot reward shaping (IQM: 0.815 vs. 0.777). These results indicate that VITA generalizes more effectively than both pre-trained (GVL) and trained (CLIP-GRU) standard sequence architectures.
>
>
> LIV has been shown to underperform on value function estimation for robotic manipulation tasks [2]. As stated in the GVL paper[2]: “GVL’s performance is also markedly better than LIV on language goals. LIV’s predictions are random, suggesting that its embedding space does not contain sufficient knowledge for predicting dense values for arbitrary unseen robot videos.” As VITA outperforms GVL on robotic manipulation and prior work has shown that LIV fails on this task domain, we implemented CLIP-GRU, which serves as a comparable sequence model baseline. We have added CLIP-GRU to Section 4 and integrated the above clarifications into the revised version.
>
>
> **W3**: Meta-World tasks can be solved with a small number of scripted demonstrations using BC.  So I'm not sure how significant the policy learning results are (it might be good to include a BC baseline, since IQL is basically weighting the data by the value fn).
>
>
>
> We thank the reviewer for allowing us to clarify the significance of the MetaWorld experiments (Section 4.5). This experiment is designed to evaluate VITA’s zero-shot reward shaping for offline RL, not to compare offline RL with imitation learning baselines (e.g., behavior cloning) on the MetaWorld MT10 benchmark. The significance of our results lies in the comparison between different reward shaping methods under the same evaluation protocol in offline RL. Notably, VITA’s zero-shot reward (IQM 0.815) outperforms VLM-based rewards (e.g., CLIP-GRU: 0.777) and the simulator’s fuzzy-logic dense reward model (Meta-WL: 0.779). We updated Section 4.5 in the revised version to clarify this.

---

> ### Author Response · Authors · 2025-11-23
> **Response to Reviewer ECyi (Part 2)**
>
> Continuing from our previous response, we address Question Q1.
>
> **Q1**: Clarification on whether TTT is applied to each latent observation within a window, with the value function predicted only after the final adapted observation.
>
>
> Thank you for raising this. VITA performs test-time training on every latent observation sequentially without resetting the adapter, following recent TTT literature for sequential data [3]. In implicit memory (VITA), updating over a local window of frames would cause the most recent observations in that window to be incorporated twice —explicitly through the window and implicitly through the accumulated parameters. In addition, updating less frequently (non-sequentially) would cause information loss because intermediate frames are omitted. Therefore, the implicit-memory test-time training paradigm requires sequential, per-timestep updates without reset.
>
>
> To empirically evaluate the window-based approach you described, we included an explicit-memory baseline (TTT-EX) in our ablation study on the impact of temporal memory (section 4.6.2 and appendix F.2).  TTT-EX performs a batched gradient update over a local window of recent frames and resets the adapter at every step. Resetting is required; otherwise, the most recent observations in the local window would be incorporated twice—explicitly through the window and implicitly through the accumulated parameters. We selected a window size of r=8 to match the window size used during VITA’s meta-training, and we used a learning rate of 1.0 with a single adaptation step, following the hyperparameter sweeps described in Appendix D.
>
>
> Our results (Appendix F.2) show that VITA outperforms TTT-EX across all environments, underscoring the importance of implicit over explicit memory for effective value function estimation in real-world robotic manipulation tasks. In the revised manuscript, we incorporated TTT-EX as an additional baseline in the ablation study (Section 4.6.2), and included the results and analysis in Appendix F.2.
>
>
> *References:*
>
> [1] Ma et al., LIV: Language-Image Representations and Rewards for Robotic Control, ICLR 2023
>
> [2] Ma et al., Vision Language Models Are In-Context Value Learners, ICLR 2025
>
> [3] Sun et al., Learning to (Learn at Test Time): RNNs with Expressive Hidden States, ICML 2025

---

> > ### Comment · Reviewer_ECyi · 2025-11-27
> >
> > Thank you for your response -- I believe several improvements to the paper have been made.
> >
> > One clarification regarding LIV -- I did not suggest using LIV zero-shot (I would expect that to underperform) -- I simply meant that because TTT is training on the data, a value function baseline that additionally trains on the target data would be reasonable. I recommended LIV because it is also CLIP based, and could be further trained on the target domain.

---

> ### Author Response · Authors · 2025-11-29
> **Response to Reviewer ECyi**
>
> We thank Reviewer ECyi for acknowledging the improvements in our work and for clarifying their suggestion to include a baseline trained on the target-domain data (W2).
>
> The CLIP-GRU baseline effectively incorporates this suggestion, as it provides a CLIP-based architecture trained on the target-domain data with temporal memory. In particular, CLIP-GRU learns to encode temporal history in recurrent hidden states, whereas a fine-tuned LIV model (LIV-FT) lacks an explicit temporal memory mechanism.
>
> In addition, since VITA is agnostic to the underlying vision–language representations, it could enhance the generalization and temporal reasoning of any pre-trained multimodal representation that encodes a universal value function (e.g., LIV, LIV-FT) via meta-learned test-time adaptation. We reiterated this point in the Related Work (Section 5.1) and Limitations and Future Work (Section 6.2) sections of the updated manuscript.

---

### Official Review · Reviewer_DuEc · 2025-11-01

**Soundness:** 3
**Presentation:** 2
**Contribution:** 3
**Rating:** 6
**Confidence:** 3

**Summary:**

This paper proposes VITA, a novel method for zero-shot goal-conditioned value function learning through test-time adaptation of vision-language models. The key contributions include: (1) A test-time adaptation framework that enhances both generalization and temporal reasoning capabilities of contrastive VLMs; (2) A dissimilarity-based sampling strategy to mitigate shortcut learning; (3) Comprehensive evaluation showing state-of-the-art performance on real-world robotic manipulation tasks under various distribution shifts, and successful application to offline RL reward shaping in simulated environments.

**Strengths:**

1. The integration of test-time adaptation for temporal reasoning in zero-shot value functions is novel and creative;
2. Extensive experiments across real-world and simulated environments demonstrate robust performance;
3. Addresses important limitations of current VLM-based value functions and enables better generalization without task-specific fine-tuning.

**Weaknesses:**

1. The computational overhead of per-timestep adaptation may limit real-time applicability in some robotic systems;
2. Lack of comparison with model baselines possessing explicit memory capabilities.

**Questions:**

The authors claim that their method has the advantage of "implicit memory through parameter updates." Could we propose the most direct comparative baseline, which would be using an explicitly memory-equipped network (e.g., RNN, Transformers) to explicitly form parameterized memory?

---

> ### Author Response · Authors · 2025-11-23
> **Response to Reviewer DuEc**
>
> We thank reviewer DUeC for the thorough feedback, the positive review, and for recognizing both the novelty of our work and the strength of our real-world and simulated experiments. As the reviewer noted, our main contribution is a test-time adaptation method that improves both generalization and temporal reasoning of VLM models used for zero-shot goal-conditioned value function estimation, addressing key limitations of current VLM-based value estimators. We address each weakness (W) and question (Q) below.
>
> **W1**: Computational overhead.
>
> In our setup, the inference-time overhead of our per-timestep adaptation is negligible. Only the adaptation module $f_{\text{adapt}}$ is updated at test time, and this update consists of a single gradient step on the learned self-supervised loss $\ell_{\text{self}}$. Since $f_{\text{adapt}}$ is a two-layer residual MLP with GELU activation and a projection dimension $d' = 64$, and we apply one gradient step per timestep, the additional computational cost is minimal and does not affect real-time applicability. We clarified this point in the discussion.
>
>
> **W2/Q1**: Lack of comparison with model baselines possessing explicit memory capabilities. Could you propose the most direct comparative baseline, which would be using an explicitly memory-equipped network (e.g., RNN, Transformers) to explicitly form parameterized memory?**
>
> We thank the reviewer for this suggestion. Explicitly memory-equipped networks encode temporal history in hidden states (e.g, RNN, GRU) or cached key–value pairs (Transformers), while keeping model parameters fixed during inference. In contrast, VITA updates its parameters sequentially at test time, so temporal history is implicitly encoded in the adapted parameters.
>
> Our baseline GVL uses a large-scale autoregressive Transformer to explicitly encode history in activation states. To provide a direct comparison with a trained network equipped with explicit memory, we implemented CLIP-GRU. This baseline follows VITA’s architecture (frozen CLIP encoder and regression head $h$), but replaces the test-time adaptation module $f_{\text{adapt}}$ with a Gated Recurrent Unit (GRU) that explicitly encodes temporal history in recurrent hidden states.
>
> VITA outperforms CLIP-GRU on progress estimation for robotic manipulation tasks for the majority of evaluation datasets (6 out of 10), particularly on the long-horizon task ms_sweep. VITA is more effective at differentiating expert from non-expert demonstrations (BinVOC: VITA 1.00 vs. CLIP-GRU 0.80). In the Meta-World MT10 benchmark, VITA outperforms CLIP-GRU for zero-shot reward shaping (IQM: VITA 0.815 vs. CLIP-GRU 0.777). These results indicate that VITA generalizes more effectively than both pre-trained (GVL) and trained (CLIP-GRU) networks equipped with explicit memory. We have added the CLIP-GRU baseline and this analysis to Section 4 of the revised manuscript.

---

### Official Review · Reviewer_M6nz · 2025-11-02

**Soundness:** 3
**Presentation:** 2
**Contribution:** 2
**Rating:** 4
**Confidence:** 4

**Summary:**

This paper introduces VITA, a zero-shot goal-conditioned value function learning framework designed to enhance generalization and temporal reasoning through test-time adaptation (TTT).
The core idea is to integrate a lightweight adaptation module that is updated online via a meta-learned self-supervised loss. Each adaptation step refines value estimation, enabling the model to implicitly capture temporal dependencies within its parameters. To avoid shortcut learning, the authors propose a dissimilarity-based sampling strategy that encourages semantic diversity during training.

Empirically, VITA achieves state-of-the-art performance on real-world robotic manipulation tasks, outperforming both autoregressive zero-shot VLMs (e.g., GVL) and CLIP-based baselines. Moreover, on the Meta-World MT10 benchmark, VITA demonstrates effective zero-shot reward shaping, even surpassing the simulator’s fuzzy-logic dense rewards.

**Strengths:**

1. The paper proposes an implicit memory formulation that effectively incorporates temporal modeling into value estimation. In addition, the proposed dissimilarity-based sampling method mitigates overfitting to chronological shortcuts—a nontrivial improvement over prior work.

2. Experimental results demonstrate that ViTA consistently and substantially outperforms existing baselines across diverse tasks and domains.

**Weaknesses:**

1. While the proposed dissimilarity-based sampling method is conceptually interesting, the paper lacks a formal theoretical explanation or empirical justification for why this approach effectively mitigates shortcut learning. Providing analytical or visual evidence would strengthen the technical soundness of the claim.

2. Key hyperparameters (e.g., learning rate, adaptation steps) are only mentioned as “selected from a sweep,” leaving uncertainty about model sensitivity.

3. The paper does not clarify why variants such as TTT-TR (same structure, fewer updates) perform worse than baselines.

4. Some formatting inconsistencies remain (e.g., inconsistent capitalization of Lself).

**Questions:**

1. TTT Variants Explanation: In Table 5, VITA outperforms both TTT-TR and TTT-RS, despite all models sharing the same network architecture. This observation raises several questions:
1.1 Could the authors clarify why the single-trajectory update variant (TTT-TR) performs worse than the baseline?
1.2 Were all training hyperparameters (e.g., learning rate, adaptation steps, λ₍self₎) kept consistent across these variants?
1.3 Have the authors explored different adaptation frequencies (e.g., updating every n steps) to evaluate how temporal update intervals influence performance?

2. Online RL Integration: How would VITA perform in an online reinforcement learning setting, where the agent interacts with the environment continuously and receives real-time feedback? Discussing or experimenting with this setup could help demonstrate the method’s robustness and adaptability beyond offline contexts.

---

> ### Author Response · Authors · 2025-11-23
> **Response to Reviewer M6nz (Part 1)**
>
> We thank reviewer M6nz  for the detailed review and for highlighting that VITA effectively incorporates temporal modeling into value estimation through implicit memory and dissimilarity-based sampling, as well as for recognizing the strength of our experimental results. Our main contribution is a test-time adaptation method that improves both generalization and temporal reasoning of VLM models used for zero-shot goal-conditioned value function estimation. We address each weakness (W) and question (Q) in the order presented below. Due to character limits, we have split our response into two separate comments. In this first comment (Part 1), we address the Weaknesses (W1–W4). Please see the subsequent comment (Part 2) for our response to the Questions.
>
>
>
>
> **W1**: Lack of formal theoretical explanation or empirical justification for why dissimilarity-based sampling effectively mitigates shortcut learning.
>
> As the reviewer noted, a strength of our method is that the proposed dissimilarity-based sampling strategy mitigates overfitting to chronological shortcuts. We empirically validate this effect through an ablation study in Section 4.5 and Appendix E.1. These experiments show that dissimilarity-based sampling improves the value function estimator's ability to distinguish expert from non-expert demonstrations compared to full-trajectory sampling. Intuitively, the dissimilarity-based sampling strategy forms mini-batches by selecting the most visually diverse sub-trajectories in each demonstration, thereby functioning as an importance-sampling mechanism (see 3.2.2 section). A formal theoretical analysis of why diversity-based sampling reduces shortcut learning is an interesting direction for future work (see Section 6.2), but it is beyond the scope of our core contribution. In the revised version, we reiterated in Section 4.5 that the effectiveness of dissimilarity-based sampling is empirically validated.
>
>
> **W2**: Key hyperparameters (e.g., learning rate, adaptation steps) are only mentioned,  leaving uncertainty about model sensitivity.
>
> Thank you for raising this point. We initially provided a sweep over learning rates \{5.0, 1.0, 0.1, 0.01\} and adaptation steps \{1, 5, 10\} for the test-time adaptation hyperparameters in Appendix D.2. To incorporate your feedback, we have now included the full results and also added two additional learning rates, so the final sweep is \{5.0, 1.0, 0.1, 0.05, 0.01, 0.001\}. We select the configuration (learning rate = 0.1, 1 step) for our main experiments based on validation performance. These results show that both a single update with a larger learning rate (learning rate =0.1, 1 step) and multiple updates with a smaller learning rate (learning rate =0.01, 10 steps) achieve comparable validation performance under test-time training. We select the former configuration because it requires fewer adaptation steps and achieves slightly better results (0.783 vs. 0.781). We have updated the full sweep results in Table 4 in Appendix D.2 of the revised version.
>
> | Learning rate | 1 Step | 5 Steps | 10 Steps |
> | :--- | :---: | :---: | :---: |
> | 5.0 | 0.169 | 0.182 | 0.157 |
> | 1.0 | 0.299 | 0.183 | 0.181 |
> | 0.1 | **0.783** | 0.512 | 0.369 |
> | 0.05 | 0.746 | 0.670 | 0.515 |
> | 0.01 | 0.454 | **0.744** | **0.781** |
> | 0.001 | 0.225 | 0.345 | 0.453 |
>
>
> **W3/Q1.1**: No clarification why variants such as TTT-TR and TTT-RS perform worse than baselines.
>
> Thank you for raising this point. VITA outperforms the TTT-TR and TTT-RS variants because VITA’s implicit memory captures both temporal order and temporal memory, unlike TTT-TR and TTT-RS. VITA is meta-trained with sequential, per-timestep updates, learning to estimate value functions through the incremental accumulation of temporal history in its parameters. TTT-TR performs a single batched gradient update over the full trajectory that averages the loss across all frames, thereby discarding temporal order. TTT-RS (memoryless) resets the adapter at every step and adapts only to the current frame, preventing any temporal information from being carried forward. This clarification has been incorporated into the ablation study (Section 4.6.2) and the appendix (Appendix F.2) in the revised version.
>
> **W4**: Formatting inconsistencies with $\ell_{\mathrm{self}}$
>
> Thank you for raising this. In the revised version, we updated $L_{\mathrm{pred}}$ to $\ell_{\mathrm{pred}}$ and corrected inconsistencies with $\ell_{\mathrm{self}}$ , so that all losses now follow a unified notation. Thank you for helping us improve the clarity of our work.

---

> ### Author Response · Authors · 2025-11-23
> **Response to Reviewer M6nz (Part 2)**
>
> Continuing from our previous response, we address the Questions (Q1-Q2).
>
>
> **Q1.1/W3**: Could the authors clarify why the single-trajectory update variant (TTT-TR) performs worse than the baseline?
>
> This question is addressed in detail in W3. In short, TTT-TR performs a single batched gradient update that averages the loss across all frames, thereby discarding temporal order.
>
> **Q1.2**: Were all training hyperparameters kept consistent across these variants?
>
> Yes. TTT-TR, TTT-RS, and VITA all use the same architecture and hyperparameters. We performed the same hyperparameter sweeps (described in W1 and Appendix D) for TTT-RS and TTT-TR, and both achieve their best performance with a learning rate of 0.1 and 1 adaptation step. As these variants serve as ablation baselines, we did not include the full sweep tables in the main paper, but we are happy to add them to the Appendix if requested for completeness.
>
> **Q1.3**: Have the authors explored different adaptation frequencies (e.g., updating every n steps)?
>
> VITA’s test-time training paradigm (implicit memory) requires updates at every step, as the model incrementally encodes temporal history into the adapter weights, following recent TTT literature for sequential data [1]. Updating less frequently inherently causes information loss because intermediate frames are omitted. In the implicit-memory setting, updating over a local window of frames while not resetting would cause the most recent observation to be incorporated twice—explicitly through the window and implicitly through the accumulated parameters. Therefore, implicit memory requires sequential, per-timestep updates without reset.
>
> To empirically evaluate whether less frequent updates could be effective in our setting, we introduced TTT-EX (explicit memory). TTT-EX performs a batched gradient update over a local window of recent frames and resets the adapter at every step. Resetting is required; otherwise, the most recent observation would be incorporated twice—explicitly through the window and implicitly through the accumulated parameters. We selected a window size of (r=8) to match the window size used during VITA’s meta-training, and we used a learning rate of 1.0 with a single adaptation step, following the hyperparameter sweeps described in Appendix D.
>
> VITA outperforms TTT-EX across all environments (Appendix F.2), underscoring the importance of implicit over explicit memory for effective value function estimation in real-world robotic manipulation tasks. In the revised manuscript, we incorporated TTT-EX as an additional baseline alongside TTT-TR and TTT-RS in the ablation study (Section 4.6.2), and reported the results in Appendix F.2.
>
>
>
>
> **Q2:** How would VITA perform in an online reinforcement learning setting, where the agent interacts with the environment continuously and receives real-time feedback?
>
> We thank the reviewer for this suggestion. In principle, VITA is well-suited for online RL because test-time adaptation allows it to adjust its reward predictions dynamically as the agent encounters out-of-distribution states during exploration. In our setup, the inference-time overhead of our per-timestep adaptation is negligible. Only the adaptation module $f_{\text{adapt}}$ is updated at test time, and this update consists of a single gradient step on the learned self-supervised loss $\ell_{\text{self}}$. Since $f_{\text{adapt}}$ is a two-layer residual MLP with GELU activation and a projection dimension (d' = 64), and we apply one gradient step per timestep, the additional computational cost is minimal and does not affect real-time applicability. While our current work focuses on evaluating VITA on (i) zero-shot generalization under distribution shifts, (ii) effective differentiation between expert and non-expert visual trajectories, and (iii) zero-shot reward shaping for offline RL, we acknowledge that applying VITA to environments with real-time feedback is a promising direction. We updated the discussion and future work sections in the revised version accordingly.
>
> *References:*
>
> [1] Sun et al., Learning to (learn at test time): RNNs with expressive hidden states, ICML 2025.

---

### Author Response · Authors · 2025-11-26
**Author Response Summary: Additional Experiments, Clarifications, and Diagrams**

We sincerely thank the reviewers for their valuable feedback and suggestions that helped us improve our work. We revised the manuscript to address the Questions (Q) and Weaknesses (W) raised by reviewers (M6nz, DuEc, ECyi).

Below, we summarize the additional experiments, clarifications, and visualizations, with detailed responses included in our earlier comments to each reviewer.

**1. Additional Experiments**

*   **New Trained Baseline** (DuEc W2/Q1; ECyi W2):
    We trained CLIP-GRU, a baseline that learns to encode temporal history in recurrent hidden states, rather than through meta-learned test-time adaptation. VITA outperforms CLIP-GRU on expert–non-expert differentiation and zero-shot reward shaping for offline RL (Section 4), indicating that implicit memory via sequential test-time updates (VITA) generalizes more effectively than explicitly encoding temporal history in recurrent hidden states (CLIP-GRU) for zero-shot value function estimation.
*   **New Ablation Baseline** (M6nz Q1.3; ECyi Q1):
    We implemented TTT-EX, which performs a batched gradient update over a local window of recent frames and resets the adapter at every test-time step. VITA outperforms TTT-EX in progress estimation accuracy across all distribution shifts (Appendix F.2), showing the effectiveness of implicit memory via sequential parameter updates over window-based updates in our setup.
*   **Hyperparameter Sensitivity** (M6nz W2/Q1.2):
    We expanded the hyperparameter sweep for test-time training and reported the full results for VITA (Appendix D.2). For the ablation baselines (TTT-TR, TTT-RS), we report the best configuration found via the same sweep procedure, but we are happy to add the full tables to the Appendix if requested for completeness.

**2. Clarifications**

*   **TTT Variants** (M6nz W3/Q1.1):
    We clarified why VITA's implicit memory mechanism outperforms the ablation strategies TTT-TR (discards temporal order via batched updates) and TTT-RS (memoryless) (Sec 4.6.2 ).
*   **Computational Overhead** (M6nz Q2, DuEc W1):
    We clarified that the inference-time overhead of VITA's per-timestep adaptation is negligible due to the lightweight adapter module and single gradient step, enabling real-time applicability (Section 6.1).
*   **Scope** (ECyi W3; M6nz W1):
    We clarified the scope and significance of the Offline RL experiments (ECyi W3) (Section 4.5) and reiterated the empirical validation of our dissimilarity-based sampling strategy (M6nz W1) (Section 4.6.1).

**3. Visualizations and Notation**

*   **New Architecture Diagram** (ECyi W1):
    We added a new diagram (Figure 2) visualizing the test-time adaptation process in detail.
*   **Updated Overview Diagram** (ECyi W1):
    We updated the overview diagram (Figure 1) to visualize meta-learning in more detail.
*   **Notation Consistency with Loss Functions** (M6nz W4):
    We standardized the notation for all loss functions.

We welcome any further comments or questions during the remaining days of the discussion period. Thank you again for your time and valuable feedback.

---

> ### Author Response · Authors · 2025-12-03
> **Author Response Summary: Final Remarks**
>
> We thank the reviewers (M6nz, DuEc, ECyi) for their valuable feedback and the Area Chair for overseeing this process. In this work, we proposed VITA, a test-time adaptation method that enhances both generalization and temporal reasoning of VLMs for zero-shot goal-conditioned value function estimation. Reviewers acknowledged VITA as novel (DuEc) and intuitive (ECyi), noting that it overcomes key limitations of prior work (M6nz, DuEc) with strong experimental results (M6nz, DuEc, ECyi).
>
> During the rebuttal period, we significantly strengthened our manuscript by addressing all limitations and questions raised by the reviewers. In particular, as summarized in our 'Author Response Summary' (posted on November 26), we provided additional experiments (CLIP-GRU, TTT-EX, hyperparameter sweeps), clarifications (TTT variants, computational overhead, scope), and visualizations (new architecture diagram, updated overview diagram).
>
> Prior to the suspension of reviewer discussions on November 27, Reviewer ECyi explicitly acknowledged that "several improvements to the paper have been made". Reviewers M6nz and DuEc had not yet responded to our detailed rebuttals (posted on November 23), which include new experimental results and clarifications fully addressing all of the questions and limitations they raised.
>
> We respectfully ask the Area Chair to consider the substantial improvements made in response to the reviewers' feedback when making the final decision.

---

### Meta-Review · Area_Chair_aUZA · 2026-01-07

**Summary:**

This paper proposes a novel strategy for test time adaptation of VLMs to goal-conditioned value prediction. Their approach takes gradient steps on a meta-learned self-supervised loss on the inference trajectory.

**Reviewer Concerns:**

The reviewers primarily had clarification questions, which were largely addressed by the response. There were also concerns about missing baselines, which have been addressed by additional experiments. One remaining concern I have regarding performance overhead is the memory overhead of requiring backpropagation compared to inference (the authors only discuss running time overhead).

**Reviewer Scores:**

The reviewers were positive on average. The negative reviewer's concerns were partially addressed, and I believe they may have raised their score.

---

### Decision · Program_Chairs · 2026-01-26

Accept (Poster)